# NoisyTraj: Robust Trajectory Prediction with Noisy Observations

## Abstract

Trajectory prediction aims to forecast an agent's future trajectories based on its historical observed trajectories, which is a critical task for various applications such as autonomous driving, robotics, and surveillance systems. Most existing trajectory prediction methods assume that the observed trajectories collected for forecasting are clean. However, in real-world scenarios, noise is inevitably introduced into the observations due to errors from sensors, detection, and tracking processes, resulting in the collapse of the existing approaches. Therefore, it is essential to perform robust trajectory prediction based on noisy observations, which is a more practical scenario. In this paper, we propose **NoisyTraj**, a noise-agnostic approach capable of tackling the problem of trajectory prediction with arbitrary types of noisy observations. Specifically, we put forward a mutual information-based mechanism to denoise the original noisy observations. This mechanism optimizes the produced trajectories to exhibit a pattern that closely resembles the clean trajectory pattern while deviating from the noisy one. Considering that the trajectory structure may be destroyed through the only optimization of mutual information, we introduce an additional reconstruction loss to preserve the structure information of the produced observed trajectories. Moreover, we further propose a ranking loss based on the intuitive idea that prediction performance using denoised trajectories should surpass that using the original noisy observations, thereby further enhancing performance. Because NoisyTraj does not rely on any specific module tailored to particular noise distributions, it can handle arbitrary types of noise in principle. Additionally, our proposed NoisyTraj can be easily integrated into existing trajectory prediction models. Extensive experiments conducted on the ETH/UCY and Stanford Drone datasets (SDD) demonstrate that NoisyTraj significantly improves the accuracy of trajectory prediction with noisy observations, compared to the baselines.

## 1 Introduction

The objective of trajectory prediction is to anticipate the future trajectories for agents given their past observed trajectories, which is an essential and emerging task in numerous applications, such as autonomous driving (Phong et al., 2024; Wang et al., 2023b; Zhou et al., 2023; 2022), drones (Corbetta et al., 2019), surveillance systems (Valera & Velastin, 2005), and robotics (Jetchev & Toussaint, 2009; Rösmann et al., 2017). In recent years, trajectory prediction has garnered significant attention in the computer vision and machine learning communities, with numerous methods proposed (Bae et al., 2023; Choi et al., 2023; Chen et al., 2023b;a). Among these methods, they typically assume the observed historical trajectories are clean, and leverage them to predict future trajectories. Recent advances have demonstrated promising performance in trajectory prediction by learning from such clean observed trajectory data.

However, in real-world scenarios, the acquisition of trajectory data is inevitably accompanied by the introduction of noise. For instance, in an autonomous driving system's trajectory acquisition pipeline, object detection is initially performed to determine the positions and categories of objects. This process is susceptible to noise stemming from sensor errors (e.g., cameras or LiDARs) or inaccuracies in the detection algorithm. Subsequently, object tracking algorithms are employed to associate the same object across multiple timestamps, thereby forming trajectories. At this stage, noise may be introduced due to occlusions and inherent errors in the tracking algorithms.

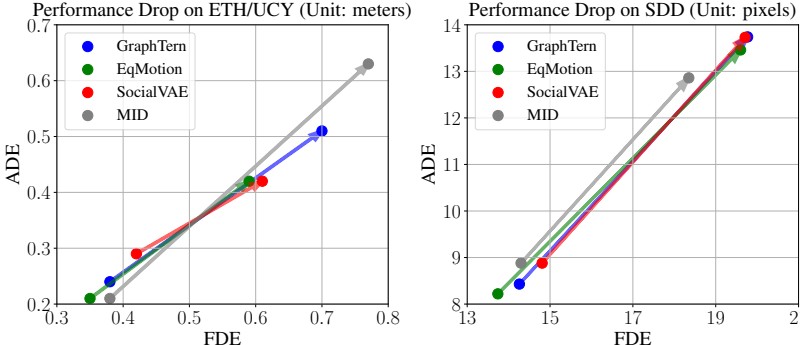

Figure 1: Performance drop of various trajectory prediction methods, including GraphTern (Bae & Jeon, 2023), EqMotion (Xu et al., 2023a), SocialVAE (Xu et al., 2022b) and MID (Gu et al., 2022), on the ETH/UCY (Pellegrini et al., 2009; Leal-Taixé et al., 2014) and Stanford Drone datasets (SDD) (Robicquet et al., 2016). The start of each arrow indicates the performance under clean observations, while the end represents the degraded performance under noisy observations (We add Gaussian noise $\mathcal{N}(0, \sigma = 0.4)$ to the clean observations.). Best viewed in color.

The presence of noise in the observed trajectories significantly hinders the performance of existing trajectory prediction methods. To substantiate this point, we corrupt the input observed trajectories by adding noise in both the training and testing stages. We then conduct experiments on the ETH/UCY (Pellegrini et al., 2009; Leal-Taixé et al., 2014) and Stanford Drone datasets (SDD) (Robicquet et al., 2016) using several recently proposed trajectory prediction approaches. We compare the performance before and after introducing noise to the observations. As shown in Figure 1, the presence of noise in the observed trajectories leads to a significant performance drop for various trajectory prediction methods on both the ETH/UCY and SDD datasets. Specifically, on the ETH/UCY dataset, the Final Displacement Error (FDE) increases from approximately 0.35-0.45 meters in the clean observation setting to 0.60-0.75 meters in the noisy observation setting. Similarly, on the SDD dataset, the FDE rises from around 13-15 pixels to 18-20 pixels This significant performance degradation highlights the detrimental impact of noise on trajectory prediction accuracy, even for state-of-the-art models. Therefore, it is crucial to devise a robust method for predicting future trajectories based on noisy observations.

In this paper, we propose **NoisyTraj**, a noise-agnostic method designed to address the challenge of trajectory prediction with arbitrary types of noisy observations. Specifically, we first propose a mutual information-based mechanism to filter noise from the original observations. This mechanism ensures the produced trajectories closely resemble the patterns of noise-free trajectories while deviating from the noisy patterns. To this end, we maximize the mutual information between the produced trajectories and the clean future trajectories (i.e., ground-truth), while simultaneously minimizing the mutual information between the produced trajectories and the original noisy observations. In this way, the produced trajectories are forced to collate information from both the noisy trajectories and clean future trajectories, thereby preserving the necessary information while filtering out noise. However, solely relying on optimizing mutual information for denoising may disrupt the structure of the trajectory. Therefore, we propose to randomly mask several observations and attempt to reconstruct the masked locations. By jointly optimizing the mutual information and reconstruction losses, the trajectory denoise model can effectively eliminate noise while preserving the structure information of the trajectory. In the meantime, we design a ranking loss to facilitate the ability of the trajectory prediction module based on an intuitive thought: predictions using the produced denoised observations will be superior to those using noisy observations. It is noteworthy that the ranking loss optimizes not only the trajectory prediction module but also the denoising module, which can further assist in filtering noise to some extent. Since NoisyTraj does not rely on any specific module tailored to a particular noise distribution, it can handle arbitrary noise in principle. Essentially, our proposed NoisyTraj is a plug-and-play approach that is compatible with existing trajectory prediction models, enabling them to gracefully handle cases with noisy observations.

Our main contributions are summarized as follows: 1) We investigate a new problem setting for trajectory prediction with noisy observations, addressing a more practical scenario. To tackle this, we propose a noise-agnostic, plug-and-play approach called NoisyTraj. 2) We design a denoising module that incorporates a mutual information-based loss along with a reconstruction loss, effectively denoising observed trajectories while preserving their structural information. 3) We propose a ranking

loss to ensure that denoised observations yield superior future predictions compared to their noisy counterparts, thereby enhancing the accuracy of trajectory predictions. 4) We conduct extensive experiments on the ETH/UCY and SDD datasets, demonstrating that our method significantly outperforms the baselines in predicting trajectory with noisy observations.

## 2 RELATED WORKS

### 2.1 TRAJECTORY PREDICTION WITH CLEAN OBSERVATIONS

Trajectory prediction has been an active area of research in the computer vision and machine learning communities. Early works employ physics-based methods to model the trajectories of agents (Luber et al., 2010; Pellegrini et al., 2009). Subsequently, learning-based approaches are proposed, which significantly enhance the performance of trajectory prediction (Zhu et al., 2023a; Rowe et al., 2023; Wang et al., 2023a; Xu et al., 2023b). They model trajectory temporal information and the interaction between agents (Alahi et al., 2016; Altché & de La Fortelle, 2017; Shi et al., 2022; Xue et al., 2018; Mohamed et al., 2020). One representative approach is social pooling, which aggregates hidden state information of neighbors within a spatial grid (Gupta et al., 2018; Sadeghian et al., 2019). Additionally, attention mechanisms (Fernando et al., 2018; Vemula et al., 2018), graph neural networks (Li et al., 2019; Kosaraju et al., 2019; Sun et al., 2020b) and transformers (Yuan et al., 2021; Zhu et al., 2023b; Shi et al., 2023) have also been exploited to model interactions among agents. To further enhance prediction performance, researchers delve into incorporating the map information. Works such as (Shafiee et al., 2021; Dendorfer et al., 2021; Sun et al., 2020a; Mangalam et al., 2021; Meng et al., 2022) encode RGB scene information, while (Ye et al., 2021; Gu et al., 2021; Zhao et al., 2021; Kang et al., 2024) incorporate lane and road traffic information. Moreover, due to the inherent uncertainty associated with agents, researchers have proposed a series of models to predict multiple plausible future trajectories, including GANs (Liang et al., 2021; Li, 2019; Zhao et al., 2019), VAEs (Lee et al., 2022; 2017; Sun et al., 2021), and diffusion models (Rempe et al., 2023; Li et al., 2024b; Jiang et al., 2023). Recently, several new task settings have been introduced to address more practical trajectory prediction problems, including momentary trajectory prediction (Li et al., 2024a; Monti et al., 2022; Sun et al., 2022; Li et al., 2024b), long-tailed distribution in trajectory prediction (Zhang et al., 2024; Mercurius et al., 2024; Wang et al., 2023c), and distribution shift in trajectory prediction (Stoler et al., 2023; Xu et al., 2022c; Kong et al., 2024).

Despite these methods having shown promising performance, they rely on sufficiently clean observed trajectories. As aforementioned, when the observed trajectories are corrupted by noise, the model performance severely deteriorates. In contrast to these approaches, we attempt to tackle the problem of predicting future trajectories with noisy observed trajectories.

### 2.2 TRAJECTORY ANOMALY DETECTION

The goal of trajectory anomaly detection is to identify abnormal patterns in trajectories, such as abnormal deviations, trajectory repetitions, and missing segments. Trajectory anomaly detection methods can be categorized as supervised learning, semi-supervised learning, and unsupervised learning approaches (Fan et al., 2009; Quispe-Torres et al., 2021; Zhang et al., 2018; Sillito & Fisher, 2008; Chebiyyam et al., 2018; Jiao et al., 2023; Mondal et al., 2021; Liatsikou et al., 2021). Supervised anomaly detection entails training a deep supervised binary or multi-class classifier using labeled data of both normal and anomalous trajectories. For instance, the work in (Chebiyyam et al., 2018) extracts statistical features from trajectories to train a multi-class SVM for classifying trajectories as normal or anomalous. Despite their promising performance, supervised methods require substantial effort to label trajectory data. To mitigate this labeling burden, researchers have explored semi-supervised anomaly detection, where only normal trajectory data are labeled. A common approach involves employing deep autoencoders trained in a semi-supervised manner (Minhas & Zelek, 2020; Song et al., 2017). These methods assume that the autoencoder will accurately encode and decode normal samples while failing to reconstruct anomalous data. Moreover, in scenarios where labeled data is scarce or unavailable, researchers have proposed unsupervised methods for trajectory anomaly detection by leveraging intrinsic data properties. Clustering is a popular unsupervised technique for trajectory anomaly detection. For example. Hu et al. (2006) clusters trajectories based on spatial and temporal information, and each motion pattern is represented with a chain of Gaussian distributions. Then, they detect anomalies based on these motion patterns. Fan et al. (Fan et al., 2009) propose

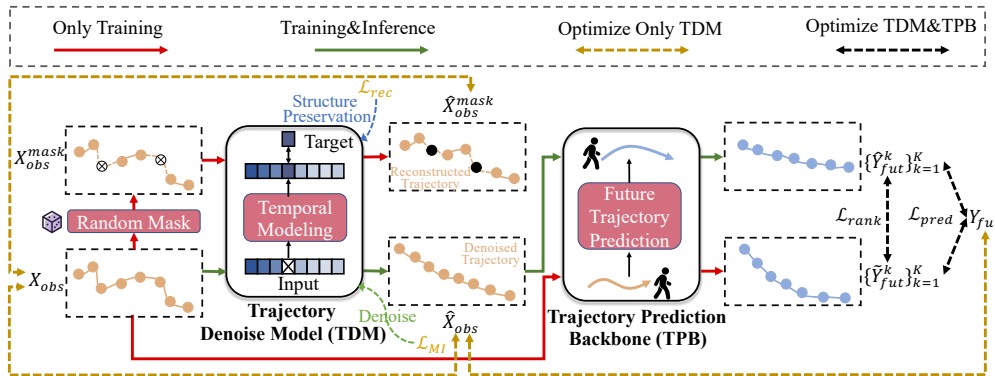

Figure 2: Overview of the proposed NoisyTraj framework. The framework is composed of two modules: a Trajectory Denoise Model (TDM) and a Trajectory Prediction Backbone (TPB). The $\mathcal{L}_{MI}$ denoises the produced trajectories $\hat{X}_{obs}$ by maximizing mutual information between $\hat{X}_{obs}$ and the clean ground-truth $Y_{fut}$, while minimizing the mutual information between $\hat{X}$ and noisy observations $X_{obs}$. The $\mathcal{L}_{rec}$ reconstructs the masked location of the original trajectories. By jointly optimizing $\mathcal{L}_{rec}$ and $\mathcal{L}_{MI}$, the trajectory denoise model can learn to denoise trajectories while preserving the structure information. The ranking loss $\mathcal{L}_{rank}$ constrains the predictions based on the denoised observations $\hat{X}_{obs}$ to be superior to those based on the noisy observations $X_{obs}$, thereby further filtering out noise and enhance the ability of trajectory prediction.

to represent trajectories with Hidden Markov Models (HMMs) and propose a dynamic hierarchical clustering method to differential normal and abnormal patterns.

While trajectory anomaly detection can identify abnormal trajectories, they cannot usually correct these trajectories. In contrast to these methods, our objective is to eliminate noise from observed trajectories and enhance the performance of trajectory prediction.

## 3 METHODS

### 3.1 PROBLEM FORMULATION

Let $X_{obs} = \{x_{obs}^1, x_{obs}^2, \ldots, x_{obs}^{T_{obs}}\}$ denote the observed trajectories, where $T_{obs}$ is the observation length, and $x_{obs}^i \in \mathbb{R}^2$ is the $i^{th}$ location. We assume the observations are given by $X_{obs} = S_{obs} + N$, where $S_{obs} = \{s_{obs}^1, s_{obs}^2, \ldots, s_{obs}^{T_{obs}}\}$ are clean observed trajectories, and $N$ is noise sample from an arbitrary distribution, such as Gaussian noise and Possion noise. Moreover, we denote the ground-truth future trajectories as $Y_{fut} = \{y_{fut}^1, y_{fut}^2, \ldots, y_{fut}^{T_{fut}}\}$, where $y_{fut}^i \in \mathbb{R}^2$ represents the $i^{th}$ locations, and $T_{fut}$ is the length of the ground-truth future trajectories. In this work, we simplify the problem by assuming $Y_{fut}$ are clean, and only the observed trajectory is noisy, which is a reasonable assumption [1]. Different from previous works that typically utilize clean observations $S_{obs}$ for future trajectory prediction, our goal is to develop a robust trajectory prediction method using noisy observations, which is a more practical scenario. Specifically, we aim to use noisy observations $X_{obs}$ to forecast $K$ plausible future trajectories $\{\hat{Y}_{fut}\}_{k=1}^K$ under the supervision of $Y_{fut}$.

### 3.2 OVERALL FRAMEWORK

The overall framework of the proposed NoisyTraj is shown in Figure 2. Our framework consists of two parts: a Trajectory Denoise Model (TDM) and a Trajectory Prediction Backbone (TPB). To eliminate noise from the noisy observations $X_{obs}$, we first propose a mutual information-based mechanism, which encourages the produced trajectories $\hat{X}_{obs}$ exhibit patterns similar to the noise-

---

[1]In practical scenarios, we can use an autonomous vehicle equipped with both cameras and LiDAR to collect training data. We can treat camera-derived trajectories as noisy data and LiDAR-derived trajectories as clean ground-truth for training. Once the model is learned based on the training data, we can deploy it on a vehicle equipped with only cameras for trajectory prediction with noisy observations.

free ground-truth future trajectories $Y_{fut}$, while deviating from the patterns of the noisy observed trajectories $X_{obs}$. This is achieved through a loss function $\mathcal{L}_{MI}$ that simultaneously maximizes the mutual information between $\hat{X}_{obs}$ and $Y_{fut}$ while minimizing the mutual information between $\hat{X}_{obs}$ and $X_{obs}$. In this way, the produced trajectories $\hat{X}_{obs}$ are forced to collate information from both the noisy trajectories $X_{obs}$ and clean future trajectories $Y_{fut}$, thereby filtering out noise. Given that only optimizing mutual information-based loss potentially disrupts the structure of the trajectories, we propose a reconstruction strategy to mitigate this issue. Specifically, we randomly mask $M$ locations of the noisy observations $X_{obs}$ to obtain $X_{obs}^{mask}$. The $X_{obs}^{mask}$ is then fed into the TDM to reconstruct the masked portion of the original noisy input $X_{obs}$ using $\mathcal{L}_{rec}$. By jointly optimizing $\mathcal{L}_{MI}$ and $\mathcal{L}_{rec}$, the TDM is able to learn to denoise while preserving the structure information of the trajectories. To facilitate more accurate trajectory prediction, we devise a ranking loss. We first input both the denoised observations $\hat{X}_{obs}$ and the original noisy observations $X_{obs}$ into the TPB to forecast future trajectories. Then the ranking loss is applied to encourage the future trajectories predicted from the denoised observations to be more precise than those predicted from the noisy observations, thereby enhancing the trajectory prediction performance. The TDM and TPB modules can benefit from each other: the ranking loss in TPB helps TDM filter noise more effectively, while the denoised trajectories generated by TDM enable TPB to predict future trajectories more accurately. As NoisyTraj is not dependent on any module specialized for a particular noise distribution, it is capable of handling arbitrary noise in principle. In addition, NoisyTraj is essentially a plug-and-play approach and can be readily integrated into existing trajectory prediction models, enabling them to effectively handle scenarios with noisy observations.

### 3.3 Trajectory Prediction with Noisy Observations

In this section, we introduce the details of our NoisyTraj. We first present the mutual information-based denoising mechanism, followed by the designed ranking loss.

#### 3.3.1 Mutual Information-Based Denoising Mechanism.

Given noisy observations $X_{obs}$, we expect the noise can be eliminated through a trajectory denoising model $\Phi_{\textbf{TDM}}$. Inspired by Information Bottleneck (Tishby et al., 2000), we encourage the produced trajectories to exhibit patterns closely resembling noise-free trajectory patterns while deviating from noisy patterns. This is achieved by maximizing the mutual information between the produced trajectories and noise-free ground-truth future trajectories $Y_{fut}$ while minimizing the mutual information between the produced trajectories and the original noisy observations $X_{obs}$. We define the objective function as:

$$J_{MI} = \min_{\hat{X}_{obs}} \alpha I(X_{obs}; \hat{X}_{obs}) - I(\hat{X}_{obs}; Y_{fut}), \tag{1}$$

where $I(\cdot \ ; \ \cdot)$ represents the mutual information and $\alpha$ is a trade-off parameter.

However, directly calculating $J_{MI}$ is intractable. Therefore, we estimate the upper bound of $I(X_{obs}; \hat{X}_{obs})$ by utilizing CLUB (Cheng et al., 2020), and the lower bound of $I(\hat{X}_{obs}; Y_{fut})$ by leveraging the method described in MINE (Belghazi et al., 2018). We first calculate the upper bound of $I(X_{obs}; \hat{X}_{obs})$.

**Theorem 3.1.** *Given two random variables $x$ and $y$, the mutual information $I(x; y)$ has the following upper bound*

$$I(x; y) \leq \mathbb{E}_{p(x,y)}[\log p(y|x)] - \mathbb{E}_{p(x)}\mathbb{E}_{p(y)}[\log p(y|x)]. \tag{2}$$

*Proof.* See proof in the Appendix 6.3.

By substituting $X_{obs}$ and $\hat{X}_{obs}$ to the Equation (2), we can obtain the upper bound of $I(X_{obs}; \hat{X}_{obs})$:

$$I(X_{obs}; \hat{X}_{obs}) \leq \mathbb{E}_{p(X_{obs}, \hat{X}_{obs})}[\log p(\hat{X}_{obs}|X_{obs})] - \mathbb{E}_{p(X_{obs})}\mathbb{E}_{p(\hat{X}_{obs})}[\log p(\hat{X}_{obs}|X_{obs})]. \tag{3}$$

Since $p(\hat{X}_{obs}|X_{obs})$ is unknown, we introduce a variational approximation distribution $q_\phi(\hat{X}_{obs}|X_{obs})$ to approximate $p(\hat{X}_{obs}|X_{obs})$ with parameter $\phi$, following (Cheng et al., 2020).

Thus, the upper bound can be written as:

$$I(X_{obs}; \hat{X}_{obs}) \leq I_\mu(X_{obs}; \hat{X}_{obs})$$
$$= \mathbb{E}_{p(X_{obs}, \hat{X}_{obs})}[\log q_\phi(\hat{X}_{obs}|X_{obs})] - \mathbb{E}_{p(X_{obs})}\mathbb{E}_{p(\hat{X}_{obs})}[\log q_\phi(\hat{X}_{obs}|X_{obs})]. \quad (4)$$

Next, we calculate the lower bound of the mutual information $I(\hat{X}_{obs}; Y_{fut})$.

**Theorem 3.2** (Donsker-Varadhan representation (Donsker & Varadhan, 1983)). *Given two probability distributions* $\mathbb{P}$, $\mathbb{Q}$. *The Kullback Liebler Divergence admits the following dual representation:*

$$D_{KL}(\mathbb{P}||\mathbb{Q}) = \sup_{T:\Omega \to \mathbb{R}} \mathbb{E}_\mathbb{P}[T] - \log \mathbb{E}_\mathbb{Q}[e^T], \quad (5)$$

*Proof.* See the proof in the Appendix 6.4.

Based on the Theorem 3.2, we can obtain the $I(\hat{X}_{obs}, Y_{fut})$ by:

$$I(\hat{X}_{obs}, Y_{fut}) = D_{KL}(p(\hat{X}_{obs}, Y_{fut})||p(\hat{X}_{obs})p(Y_{fut})) \quad (6)$$
$$= \sup_{T:\Omega \to \mathbb{R}} \mathbb{E}_{p(\hat{X}_{obs}, Y_{fut})}[T] - \log \mathbb{E}_{p(\hat{X}_{obs})p(Y_{fut})}[e^T], \quad (7)$$

where $\Omega = \hat{X}_{obs} \times Y_{fut}$ is the input space. Let $\mathcal{F}$ be any class of functions $T : \Omega \to \mathbb{R}$, and the lower bound of $I(\hat{X}_{obs}, Y_{fut})$ can be expressed as:

$$I(\hat{X}_{obs}, Y_{fut}) \geq I_\mathcal{F}(\hat{X}_{obs}, Y_{fut}) = \sup_{T \in \mathcal{F}} \mathbb{E}_{p(\hat{X}_{obs}, Y_{fut})}[T] - \log \mathbb{E}_{p(\hat{X}_{obs})p(Y_{fut})}[e^T]. \quad (8)$$

We choose $\mathcal{F}$ to be the family of functions $T_\psi : \hat{X}_{obs} \times Y_{fut} \to \mathbb{R}$, parameterized by a neural network $\psi$. Thus, the lower bound can be written as:

$$I(\hat{X}_{obs}, Y_{fut}) \geq I_\psi(\hat{X}_{obs}, Y_{fut}) = \sup_\psi \mathbb{E}_{p(\hat{X}_{obs}, Y_{fut})}[T_\psi] - \log \mathbb{E}_{p(\hat{X}_{obs})p(Y_{fut})}[e^{T_\psi}]. \quad (9)$$

Based on Equation (4) and (9), we derive the upper bound $\mathcal{L}_{MI}$ of the $J_{MI}$ as:

$$J_{MI} \leq \mathcal{L}_{MI} = \alpha I_\mu(X_{obs}; \hat{X}_{obs}) - I_\psi(\hat{X}_{obs}, Y_{fut})$$
$$= \alpha \mathbb{E}_{p(X_{obs}, \hat{X}_{obs})}[\log q_\phi(\hat{X}_{obs}|X_{obs})] - \mathbb{E}_{p(X_{obs})}\mathbb{E}_{p(\hat{X}_{obs})}[\log q_\phi(\hat{X}_{obs}|X_{obs})]$$
$$- \sup_\psi \mathbb{E}_{p(\hat{X}_{obs}, Y_{fut})}[T_\psi] + \log \mathbb{E}_{p(\hat{X}_{obs})p(Y_{fut})}[e^{T_\psi}]. \quad (10)$$

By minimizing the upper bound $\mathcal{L}_{MI}$, we can obtain an approximation solution to Equation (1), enabling the trajectory denoise model to learn how to denoise trajectories.

However, only optimizing the mutual information may destroy the structure of the produced trajectories. Therefore, we propose a reconstruction strategy to preserve the structure information. As shown in the left part of Figure 2, we mask locations within the noisy observed trajectories $X_{obs}$ to generate $X_{obs}^{mask}$, which can be formulated as:

$$X_{obs}^{mask} = X_{obs} \odot \mathcal{M}_{obs}, \quad (11)$$

where $\odot$ represents the element-wise multiplication. $\mathcal{M}_{obs}$ is a 0-1 mask vector, where the value **0** represents the corresponding locations are masked. Subsequently, the $X_{obs}^{mask}$ is fed into the trajectory denoise model $\Phi_{\textbf{TDM}}$ to produce observed trajectories $\hat{X}_{obs}^{mask}$. To enable TDM to preserve the structural information of the trajectories, we reconstruct the masked locations in the produced trajectories. We define the reconstruction loss as follows:

$$\mathcal{L}_{rec} = \mathcal{J}(\hat{X}_{obs}^{mask} \odot (1 - \mathcal{M}_{obs}), X_{obs} \odot (1 - \mathcal{M}_{obs})), \quad (12)$$

where $\mathcal{J}$ denotes the distance metric, and we empirically adopt $L_2$ distance in this work. Through optimizing $\mathcal{L}_{rec}$, the trajectory denoise model can learn to preserve the structure information of the observations. By jointly optimizing the reconstruction loss $\mathcal{L}_{rec}$ with $\mathcal{L}_{MI}$, the mutual information-based mechanism effectively denoises the trajectories while preserving their structural information.

### 3.3.2 TRAJECTORY PREDICTION BASED ON RANKING LOSS.

After obtaining the denoised observed trajectories $\hat{X}_{obs}$, we design a ranking loss to enhance future trajectory prediction performance. The ranking loss is based on an intuitive thought: leveraging denoised observed trajectories should yield more accurate future predictions compared to using noisy observations. To accomplish this, we first input the denoised observations into the trajectory prediction backbone $\Phi_{\textbf{TPB}}$. Then, we can predict $K$ plausible future trajectories:

$$\{\hat{Y}_{fut}^k\}_{k=1}^K = \Phi_{\textbf{TPB}}(\Phi_{\textbf{TDM}}(X_{obs})), \tag{13}$$

Similarly, we can also predict $K$ possible trajectories based on the noisy observed trajectories:

$$\{\tilde{Y}_{fut}^k\}_{k=1}^K = \Phi_{\textbf{TPB}}(X_{obs}). \tag{14}$$

After obtaining $K$ possible trajectories based on the denoised and noisy observed trajectories, respectively, we then select the minimal distances $d_{denoise}$ and $d_{noise}$ by calculating the distances between each predicted trajectory and ground-truth trajectory, respectively. Formally,

$$d_{denoise} = \min_{1 \le k \le K} ||\hat{Y}_{fut}^k - Y_{fut}||_2, \;\; d_{noise} = \min_{1 \le k \le K} ||\tilde{Y}_{fut}^k - Y_{fut}||_2. \tag{15}$$

Then, we employ the ground-truth future trajectories as supervision for the best-predicted trajectory:

$$\mathcal{L}_{pred} = ||\hat{Y}_{fut}^{best} - Y_{fut}||_2 + ||\tilde{Y}_{fut}^{best} - Y_{fut}||_2, \tag{16}$$

where $best$ represents the trajectory with a minimal distance to the ground-truth. Subsequently, we design a ranking loss to constrain the best prediction $\hat{Y}_{fut}^{best}$ using the denoised observations to be more accurate than that $\tilde{Y}_{fut}^{best}$ using the noisy observations:

$$\mathcal{L}_{rank} = \max(0, d_{denoise} - d_{noise} + \Delta), \tag{17}$$

where $\Delta$ is a margin. Since the ranking loss optimizes both the trajectory prediction backbone and the trajectory denoise model, it not only aids in better trajectory prediction but also facilitates the denoising ability of the trajectory denoise model.

### 3.4 OPTIMIZATION AND INFERENCE

**Optimization.** We define the total loss function as:

$$\mathcal{L} = \mathcal{L}_{pred} + \beta \mathcal{L}_{rank} + \delta \mathcal{L}_{rec} + \gamma \mathcal{L}_{MI}, \tag{18}$$

where $\beta$, $\delta$, and $\gamma$ are trade-off hyperparameters. The training details are shown in Appendix 6.7.

**Inference.** After training, the model can be utilized for trajectory prediction based on noisy observations. As shown in the blue arrow in Figure 2, we first feed the noisy trajectories $X_{obs}$ into the trajectory denoise model $\Phi_{\textbf{TDM}}$ to obtain denoised trajectories $\hat{X}_{obs}$. Subsequently, the trajectory prediction backbone $\Phi_{\textbf{TPB}}$ takes $\hat{X}_{obs}$ as input to obtain the predicted future trajectories $\{\hat{Y}_{fut}^k\}_{k=1}^K$.

## 4 EXPERIMENTS

### 4.1 EXPERIMENT SETTINGS

**Dataset.** We evaluate our proposed NoisyTraj on two widely used datasets: the ETH/UCY (Pellegrini et al., 2009; Leal-Taixé et al., 2014) and SDD dataset (Robicquet et al., 2016). The ETH/UCY dataset is composed of 5 scenes, including ETH, HOTEL, UNIV, ZARA1, and ZARA2, with 1,536 pedestrians recorded in total. Following (Huang et al., 2019; Xu et al., 2022b; Mangalam et al., 2020; Bae et al., 2023), we adopt the "leave-one-out" strategy, where the models are trained on 4 scenes and tested on the remaining scene. SDD consists of 20 scenes captured using a drone in a top-down view around the university campus containing several moving agents such as humans, bicyclists, skateboarders, and vehicles, which contains 5,232 trajectories in total. We follow a common setting among existing works, where 8 frames of trajectories (3.2 seconds) are used as observations to predict the next 12 frames (Wong et al., 2022; Gu et al., 2022). To verify the robustness of NoisyTraj against noise, we add noise into existing publicly available trajectory prediction datasets, ETH/UCY and

Table 1: Comparison of different methods on the ETH/UCY dataset. The evaluation metrics are ADE and FDE (Unit: meters). The best results are highlighted in **bold**.

| Noise | Method | ETH | | HOTEL | | UNIV | | ZARA1 | | ZARA2 | | AVG | |
|---|---|---|---|---|---|---|---|---|---|---|---|---|---|
| | | ADE | FDE | ADE | FDE | ADE | FDE | ADE | FDE | ADE | FDE | ADE | FDE |
| $\sigma = 0.2$ | GraphTern | 0.56 | 0.81 | 0.27 | 0.39 | 0.39 | 0.59 | 0.37 | 0.58 | 0.34 | 0.50 | 0.39 | 0.57 |
| | EqMotion | 0.51 | 0.70 | 0.17 | 0.24 | 0.36 | 0.56 | 0.33 | 0.54 | 0.25 | 0.37 | 0.32 | 0.48 |
| | MID | 0.74 | 0.86 | 0.37 | 0.36 | 0.45 | 0.58 | 0.43 | 0.52 | 0.40 | 0.47 | 0.48 | 0.56 |
| | SocialImplict | 0.67 | 1.28 | 0.31 | 0.49 | 0.46 | 0.77 | 0.39 | 0.71 | 0.35 | 0.63 | 0.44 | 0.78 |
| | SocialVAE | 0.56 | 0.89 | 0.18 | 0.25 | 0.40 | 0.63 | 0.32 | 0.49 | 0.25 | 0.38 | 0.34 | 0.53 |
| | Wavelet+GraphTern | 0.50 | 0.72 | 0.23 | 0.35 | 0.37 | 0.56 | 0.32 | 0.50 | 0.30 | 0.42 | 0.34 | 0.51 |
| | EMA+GraphTern | 0.53 | 0.76 | 0.26 | 0.37 | 0.38 | 0.57 | 0.34 | 0.54 | 0.31 | 0.48 | 0.36 | 0.53 |
| | **NoisyTraj+GraphTern** | **0.48** | **0.68** | **0.19** | **0.26** | **0.35** | **0.53** | **0.28** | **0.44** | **0.27** | **0.36** | **0.31** | **0.45** |
| | Wavelet+EqMotion | 0.48 | 0.64 | **0.16** | 0.22 | 0.32 | 0.52 | 0.31 | 0.48 | 0.23 | 0.32 | 0.30 | 0.44 |
| | EMA+EqMotion | 0.49 | 0.66 | **0.16** | 0.22 | 0.34 | 0.53 | 0.31 | 0.48 | 0.23 | 0.33 | 0.31 | 0.44 |
| | **NoisyTraj+EqMotion** | **0.47** | **0.61** | **0.16** | **0.21** | **0.29** | **0.47** | **0.28** | **0.44** | **0.21** | **0.29** | **0.28** | **0.40** |
| $\sigma = 0.4$ | GraphTern | 0.67 | 0.99 | 0.40 | 0.51 | 0.49 | 0.69 | 0.51 | 0.71 | 0.46 | 0.61 | 0.51 | 0.70 |
| | EqMotion | 0.65 | 0.87 | 0.25 | 0.34 | 0.43 | 0.64 | 0.42 | 0.62 | 0.34 | 0.48 | 0.42 | 0.59 |
| | MID | 0.88 | 1.08 | 0.51 | 0.49 | 0.60 | 0.73 | 0.59 | 0.84 | 0.57 | 0.72 | 0.63 | 0.77 |
| | SocialImplict | 0.74 | 1.39 | 0.40 | 0.64 | 0.54 | 0.86 | 0.59 | 0.88 | 0.53 | 0.76 | 0.56 | 0.61 |
| | SocialVAE | 0.67 | 1.01 | 0.24 | 0.32 | 0.48 | 0.73 | 0.41 | 0.58 | 0.31 | 0.43 | 0.42 | 0.61 |
| | Wavelet+GraphTern | 0.60 | 0.83 | 0.33 | 0.43 | 0.46 | 0.64 | 0.42 | 0.63 | 0.37 | 0.49 | 0.44 | 0.60 |
| | EMA+GraphTern | 0.64 | 0.92 | 0.36 | 0.46 | 0.46 | 0.66 | 0.47 | 0.67 | 0.41 | 0.58 | 0.47 | 0.66 |
| | **NoisyTraj+GraphTern** | **0.55** | **0.77** | **0.29** | **0.41** | **0.42** | **0.61** | **0.38** | **0.56** | **0.34** | **0.42** | **0.40** | **0.55** |
| | Wavelet+EqMotion | 0.62 | 0.74 | 0.23 | 0.29 | 0.41 | 0.58 | 0.39 | 0.56 | 0.32 | 0.43 | 0.39 | 0.52 |
| | EMA+EqMotion | 0.63 | 0.75 | 0.23 | 0.29 | 0.42 | 0.63 | 0.40 | 0.58 | 0.31 | 0.42 | 0.43 | 0.53 |
| | **NoisyTraj+EqMotion** | **0.57** | **0.71** | **0.20** | **0.25** | **0.35** | **0.51** | **0.35** | **0.51** | **0.29** | **0.41** | **0.35** | **0.48** |

SDD. We employ two settings: i) we first add Gaussian noise $\mathcal{N}(0, \sigma)$ based on the Central Limit Theorem (Kwak & Kim, 2017), which suggests that combination of various noise sources—such as sensor error, detection error, and tracking error—tend to approximate a Gaussian distribution. To verify the effectiveness of NoisyTraj under different levels of noise, we set $\sigma$ to different values, e.g., 0.2 and 0.4. (Forde & Daniel, 2021); ii) to verify the noise-agnostic property of NoisyTraj, we add various types of noise including Poisson noise, mixed noise, and multiplicative noise.

**Evaluation Metrics.** Following previous works (Mao et al., 2023; Gu et al., 2022; Sadeghian et al., 2019; Shi et al., 2021), we employ Average Displacement Error (ADE) and Final Displacement Error (FDE) to evaluate the predicted trajectories. ADE is the average L2 error between all future timesteps, and FDE is the error at the final timestamp. We take the best out of $K = 20$ predictions to account for the multi-modality for trajectory prediction, as in (Salzmann et al., 2020; Xu et al., 2022a).

**Backbones and Compared Baselines.** To validate the efficacy of NoisyTraj, we integrate it into two popular trajectory prediction backbones GraphTern (Bae & Jeon, 2023) and EqMotion (Xu et al., 2023a). We first compare our method against five state-of-the-art trajectory prediction models, including **GraphTern**, **EqMotion**, **MID** (Gu et al., 2022), **SocialImplict** (Mohamed et al., 2022) and **SocialVAE** (Xu et al., 2022b). These methods take original noisy observations as input to predict future trajectories. Considering there are few works focusing on trajectory prediction with noisy observations, we establish two trajectory denoising baselines by integrating Wavelet and EMA with the trajectory prediction backbones, respectively, for a more comprehensive comparison. The **Wavelet** utilizes the wavelet transform to decompose the signal into multiple scales, obtaining wavelet coefficients of different frequencies. Then the noise in high-frequency coefficients is removed by the thresholding method. The **EMA** smooths the current trajectory location by taking an exponentially weighted average of the current location and past locations.

## 4.2 RESULTS AND ANALYSIS

**Performance on Trajectory Predictions with Noisy Observations.** We evaluate the performance of our proposed NoisyTraj and compare it with various baselines on the ETH/UCY and SDD datasets. The results are listed in Table 1 and Table 2. Based on the two tables, NoisyTraj+GraphTern and NoisyTraj+EqMotion significantly outperforms GraphTern and EqMotion on the two datasets under

Table 2: Comparison of different methods on the SDD dataset. The evaluation metrics are ADE and FDE (Unit: pixels). The best results are highlighted in **bold**.

| Noise | Method | SDD ADE | FDE | Noise | Method | SDD ADE | FDE |
|---|---|---|---|---|---|---|---|
| | GraphTern | 11.67 | 18.37 | | GraphTern | 13.74 | 19.77 |
| | EqMotion | 10.62 | 15.68 | | EqMotion | 13.46 | 19.60 |
| | MID | 10.26 | 15.38 | | MID | 12.86 | 18.35 |
| | SocialImplicit | 15.92 | 26.82 | | SocialImplicit | 18.07 | 29.98 |
| | SocialVAE | 11.67 | 17.62 | | SocialVAE | 13.73 | 19.71 |
| $\sigma = 0.2$ | Wavelet+GraphTern | 10.52 | 16.86 | $\sigma = 0.4$ | Wavelet+GraphTern | 12.98 | 18.50 |
| | EMA+GraphTern | 11.03 | 17.42 | | EMA+GraphTern | 13.21 | 19.11 |
| | **NoisyTraj+GraphTern** | **10.08** | **15.64** | | **NoisyTraj+GraphTern** | **12.35** | **17.28** |
| | Wavelet+EqMotion | 10.36 | 15.24 | | Wavelet+EqMotion | 12.38 | 18.25 |
| | EMA+EqMotion | 10.32 | 15.22 | | EMA+EqMotion | 12.79 | 18.64 |
| | **NoisyTraj+EqMotion** | **10.06** | **14.67** | | **NoisyTraj+EqMotion** | **11.92** | **17.65** |

the setting of $\sigma = 0.2$ and $\sigma = 0.4$ meters. This illustrates current state-of-the-art methods cannot well tackle the case of noisy observations. However, when integrating our proposed NoisyTraj into these two models, the performance can be significantly improved. This demonstrates the effectiveness of our method for trajectory prediction with noisy observations, and also highlights its compatibility with different trajectory prediction models. Furthermore, NoisyTraj outperforms the Kalman and EMA denoising methods, further underscoring the superiority of our proposed approach.

**Ablation Studies.** We conduct ablation studies on the components of our proposed method. We utilize GraphTern as the backbone and set $\sigma$ to 0.4 meters. The results are listed in Table 3. We first incorporate the mutual information loss $\mathcal{L}_{MI}$ into GraphTern, the performance is improved, demonstrating our denoising mechanism is effective. We then add $\mathcal{L}_{rec}$ into our method to reconstruct the masked locations for preserving the structure information of the tra-

Table 3: Ablation Studies on each component of NoisyTraj. The best results are highlighted in **bold.**

| Component | | | ETH/UCY | | SDD | |
|---|---|---|---|---|---|---|
| $\mathcal{L}_{MI}$ | $\mathcal{L}_{rec}$ | $\mathcal{L}_{rank}$ | ADE | FDE | ADE | FDE |
| | | | 0.51 | 0.70 | 13.74 | 19.77 |
| ✓ | | | 0.47 | 0.65 | 13.21 | 18.79 |
| ✓ | ✓ | | 0.42 | 0.59 | 12.74 | 17.96 |
| ✓ | ✓ | ✓ | **0.40** | **0.55** | **12.35** | **17.28** |

jectories. We observe a further improvement in performance, which demonstrates its effectiveness. Finally, we add the ranking loss $\mathcal{L}_{rank}$, which enables our method to achieve the best performance. This indicates that $\mathcal{L}_{rank}$ enhances the capability of the trajectory prediction model.

Table 4: Comparison of different methods under different noise setting on the SDD dataset. The evaluation metrics are ADE and FDE (Unit: pixels). The best results are highlighted in **bold**.

(a) Possion Noise ($\lambda = 0.4$).

| Noise | Method | SDD ADE | FDE |
|---|---|---|---|
| | EqMotion | 14.05 | 19.46 |
| $\lambda = 0.4$ | Wavelet+EqMotion | 12.95 | 17.97 |
| | EMA+EqMotion | 13.15 | 17.58 |
| | **NoisyTraj+EqMotion** | **12.22** | **16.23** |

(b) Mixed noise composed of Gaussian noise ($\sigma = 0.2$) and Poisson noise ($\lambda = 0.2$).

| Noise | Method | SDD ADE | FDE |
|---|---|---|---|
| | EqMotion | 14.66 | 20.27 |
| $\sigma = 0.2$ | Wavelet+EqMotion | 13.48 | 18.30 |
| $\lambda = 0.2$ | EMA+EqMotion | 13.82 | 18.97 |
| | **NoisyTraj+EqMotion** | **12.96** | **17.09** |

(c) Noise randomly multiplied by $\delta \in [0.95, 1.0]$.

| Noise | Method | SDD ADE | FDE |
|---|---|---|---|
| | EqMotion | 17.48 | 19.10 |
| $\delta \in$ | Wavelet+EqMotion | 16.17 | 18.18 |
| $[0.95, 1.0]$ | EMA+EqMotion | 16.25 | 18.38 |
| | **NoisyTraj+EqMotion** | **15.66** | **17.39** |

(d) Gaussian Noise sampled from $\sigma \in \{0.2, 0.4\}$.

| Noise | Method | SDD ADE | FDE |
|---|---|---|---|
| | EqMotion | 13.33 | 18.92 |
| $\sigma \in$ | Wavelet+EqMotion | 12.82 | 15.66 |
| $\{0.2, 0.4\}$ | EMA+EqMotion | 12.76 | 15.50 |
| | **NoisyTraj+EqMotion** | **12.28** | **15.15** |

**Performance under Different Noise Setting.** We conduct experiments to verify the effectiveness of NoisyTraj for various noise settings. we added (1) Poisson noise, (2) Mixed noise (Gaussian + Poisson), (3) Noise randomly multiplied by $\delta \in [0.95, 1]$ and (4) Gaussian noise randomly sampled from $\sigma \in \{0.2, 0.4\}$. The results are listed in Table 4. We observe our method consistently

outperforms the baselines across various settings, which demonstrates the effectiveness of Noisytraj and it is agnostic to noise distributions in principle.

**Generalizability of NoisyTraj.** To verify the generalizability of our method, we conduct additional experiments where the noise in the training and validation/testing set is different. Specifically, we train the model using trajectories with Gaussian noise ($\sigma = 0.4$) and then test it with Gaussian noise ($\sigma = 0.2$) and Poisson noise ($\lambda = 0.4$). The results, as listed in Table 5, show that our NoisyTraj can achieve denoising effectively, and still outperforms the baselines. This also indicates that our method possesses generalization ability when noise is different in the training and testing/validation set. Therefore, we believe our method still works when facing real-world noisy trajectories.

Table 5: Comparison of different methods when noise is different between training and testing on the SDD dataset. The evaluation metrics are ADE and FDE (Unit: pixels). The best results are highlighted in **bold**.

| Noise | Method | SDD ADE | SDD FDE | Noise | Method | SDD ADE | SDD FDE |
|---|---|---|---|---|---|---|---|
| Train: $\sigma = 0.4$ Test: $\sigma = 0.2$ | EqMotion | 11.47 | 16.82 | Train: $\sigma = 0.4$ Test: $\lambda = 0.4$ | EqMotion | 15.03 | 19.35 |
| | Wavelet+EqMotion | 10.86 | 16.07 | | Wavelet+EqMotion | 14.40 | 18.56 |
| | EMA+EqMotion | 10.99 | 16.24 | | EMA+EqMotion | 14.57 | 18.98 |
| | **NoisyTraj+EqMotion** | **10.72** | **15.89** | | **NoisyTraj+EqMotion** | **14.26** | **18.32** |

**Qualitative Results.** We visualize the denoised observations and predicted future trajectories generated by EWA, Wavelet, and NoisyTraj, using GraphTern as the backbone. The results are shown in Figure 3. We observe that NoisyTraj can generate less noisy observed trajectories and more accurate future trajectories compared to other methods. This demonstrates the proposed mutual information-based mechanism effectively denoises the observations, and the ranking loss aids in forecasting more precise future trajectories.

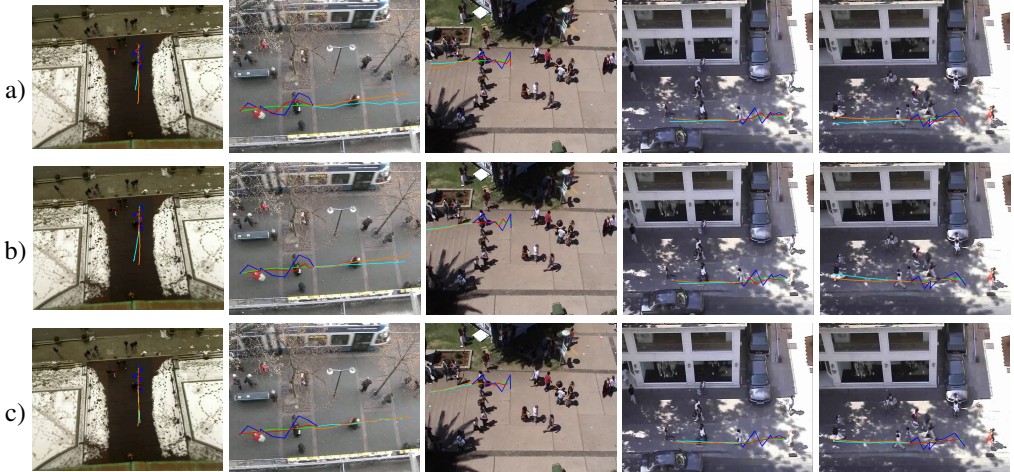

Figure 3: Visualization of predicted trajectories via (a) EMA+GraphTern, (b) Wavelet+GraphTern, (c) NoisyTraj+GraphTern on the ETH/UCY Dataset. The clean, noisy, and denoised observations are shown in green, blue, and red, respectively. The noisy observations are obtained through adding Gaussian noise $\mathcal{N}(0, \sigma = 0.4)$ into clean observations. The ground-truth and predicted future trajectories are shown in orange and cyan, respectively.

## 5 CONCLUSION

In this paper, we investigated an extremely challenging task of trajectory prediction with noisy observations. We proposed NoisyTraj, a framework that simultaneously filters out noise and predicts future trajectories, enabling them to benefit from each other. To remove the noise from the observations, we designed a denoising mechanism by jointly optimizing a mutual information-based loss and a reconstruction loss. Moreover, we devised a ranking loss that requires the prediction performance using denoised observed trajectories to be superior to that using the original noisy observations, thereby further improving the performance of the model. Extensive experiments demonstrated the effectiveness of NoisyTraj and its compatibility with various trajectory prediction models.

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

# 6 APPENDIX

## 6.1 VISUALIZATION OF MUTUAL INFORMATION-BASED DENOISING MECHANISM

To further demonstrate the efficacy of the proposed Mutual Information-Based Denoising Mechanism, we visualize the denoised trajectory and future predicted trajectory on the ETH dataset. As shown in Figure 4(a), optimizing solely for mutual information leads to the destruction of structural information. However, as depicted in the Figure 4(b), when we incorporate the reconstruction loss $\mathcal{L}_{rec}$, the structure of the trajectory is preserved, and more accurate future trajectory predictions based on these well-structured observations. This underscores the effectiveness of our proposed method.

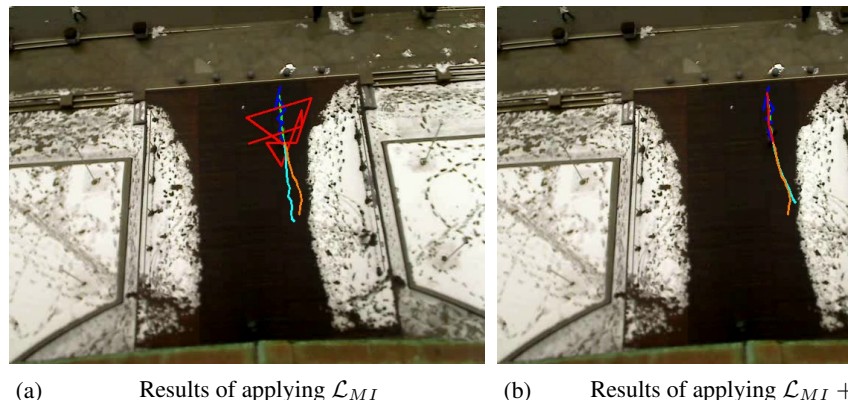

(a)    Results of applying $\mathcal{L}_{MI}$     (b)    Results of applying $\mathcal{L}_{MI} + \mathcal{L}_{rec}$

Figure 4: Visualization of trajectories on ETH dataset by employing (a) $\mathcal{L}_{MI}$ and (b) $\mathcal{L}_{MI} + \mathcal{L}_{rec}$. The clean, noisy, and denoised observations are shown in green, blue, and red, respectively. The ground-truth and predicted future trajectories are shown in orange and cyan, respectively.

## 6.2 MORE ANALYSIS OF NOISYTRAJ

**Performance under low/no noise settings.** We evaluate NoisyTraj under low or no noise by setting the Gaussian noise $\sigma$ to 0.05 and 0. The results presented in Table 6 indicate that after integrating NoisyTraj into EqMotion, the performance is still superior to baselines when at a low noise level ($\sigma = 0.05$). Additionally, NoisyTraj+EqMotion performs comparably to EqMotion when $\sigma = 0$. This demonstrates NoisyTraj does not degrade the performance when noise is not introduced.

Table 6: Comparison of different methods under different noise setting on the SDD dataset. The evaluation metrics are ADE and FDE (Unit: pixels). The best results are highlighted in **bold**.

| Noise | Method | SDD ADE | FDE |
|---|---|---|---|
| $\sigma = 0.05$ | EqMotion | 8.48 | 13.49 |
| | Wavelet+EqMotion | 8.39 | 13.37 |
| | EMA+EqMotion | 8.42 | 13.36 |
| | NoisyTraj+EqMotion | **8.32** | **13.28** |

| Noise | Method | SDD ADE | FDE |
|---|---|---|---|
| $\sigma = 0$ | EqMotion | **8.08** | 13.12 |
| | Wavelet+EqMotion | 8.16 | 13.42 |
| | EMA+EqMotion | 8.22 | 13.57 |
| | NoisyTraj+EqMotion | 8.11 | **13.08** |

Table 7: Comparison with baselines using MID backbone. The evaluation metrics are ADE and FDE (Unit: pixels). The best results are highlighted in **bold**.

| Noise | Method | SDD ADE | FDE |
|---|---|---|---|
| $\sigma = 0.4$ | MID | 12.86 | 18.35 |
| | Wavelet+MID | 12.26 | 17.88 |
| | EMA+MID | 12.45 | 18.01 |
| | NoisyTraj+MID | **11.97** | **17.41** |

**Performance on diffusion-based backbones.** In addition to GraphTern and EqMotion, we integrate NoisyTraj into MID, a diffusion-based model for trajectory prediction. Specifically, we first use

TDM to denoise the noisy observations $X_{obs}$, obtaining $\hat{X}_{obs}$ . Then, using both the denoised and original observations, we sample normal noise from a standard Gaussian distribution to generate $\hat{Y}_{fut}$ and $\tilde{Y}_{fut}$, respectively. To optimize the model, We apply $\mathcal{L}_{pred}$ and $\mathcal{L}_{rank}$ alongside the MID loss . The results shown in Table 7 show that NoisyTraj still outperforms the baselines, which further underscores its adaptability.

**Comparison with frozen predictor.** We conduct an experiment where we freeze the predictor and only train the denoiser. We first load the predictor trained on clean observations, freeze its parameters, and then integrate NoisyTraj, training only the denoiser. The results, shown in Table 8, reveal a performance decrease when the predictor's parameters are frozen. This indicates the necessity of jointly learning the denoiser and predictor.

Table 8: Comparison with NoiseTraj where the predictor is freezed. The best results are highlighted in **bold**

| Noise | Method | SDD | |
|---|---|---|---|
| | | ADE | FDE |
| | EqMotion | 13.46 | 19.60 |
| $\sigma = 0.4$ | NoisyTraj+EqMotion (freeze) | 12.19 | 17.95 |
| | NoisyTraj+EqMotion | **11.92** | **17.65** |

**Comparison with Learning-based baseline.** To our knowledge, our work is the first to address trajectory prediction with noisy observations, with no existing learning-based baselines for this problem. We use Noise2Void [1], a learning-based denoiser originally for image denoising, as another baseline. We first denoise the observed trajectories, and then perform future trajectory prediction based on the observations. The results in Table 9 of the attached PDF show that NoisyTraj outperforms Noise2Void, demonstrating the effectiveness of our method.

Table 9: Comparison with baselines on SDD dataset. The evaluation metrics are ADE and FDE (Unit: pixels). The best results are highlighted in **bold**.

| Noise | Method | SDD | |
|---|---|---|---|
| | | ADE | FDE |
| | EqMotion | 13.46 | 19.60 |
| | Wavelet+EqMotion | 12.38 | 18.25 |
| $\sigma = 0.4$ | EMA+EqMotion | 12.79 | 18.64 |
| | Noise2Void+EqMotion | 12.46 | 18.52 |
| | NoisyTraj+EqMotion | **11.92** | **17.65** |

## 6.3 PROOF OF THEOREM 3.1

**Theorem 6.1** (Theorem 3.1 restated). *Given two random variables $x$ and $y$, the mutual information $I(x; y)$ has the following upper bound*

$$I(x; y) \leq \mathbb{E}_{p(x,y)}[\log p(y|x)] - \mathbb{E}_{p(x)}\mathbb{E}_{p(y)}[\log p(y|x)] \tag{19}$$

*Proof.* The definition of mutual information between variables x and y is

$$I(x; y) = \mathbb{E}_{p(x,y)}\left[\log \frac{p(x,y)}{p(x)p(y)}\right] = \mathbb{E}_{p(x,y)}\left[\log \frac{p(y|x)}{p(y)}\right]$$
$$= \mathbb{E}_{p(x,y)}[\log p(y|x)] - \mathbb{E}_{p(x,y)}[\log p(y)]$$
$$= \mathbb{E}_{p(x,y)}[\log p(y|x)] - \mathbb{E}_{p(y)}[\log p(y)] \tag{20}$$

By the definition of the marginal distribution, we have:

$$p(y) = \int p(y|x)p(x)dx = \mathbb{E}_{p(x)}[p(y|x)]. \tag{21}$$

By substituting Equation (21) to , we have:

$$
\begin{aligned}
I(x;y) &= \mathbb{E}_{p(x,y)}[\log p(y|x)] - \mathbb{E}_{p(y)}[\log p(y)] \\
&= \mathbb{E}_{p(x,y)}[\log p(y|x)] - \mathbb{E}_{p(y)}[\log \mathbb{E}_{p(x)}[p(y|x)]]
\end{aligned}
\tag{22}
$$

Note that the $\log(\cdot)$ is a concave function, by Jensen's Inequality, we have

$$
\begin{aligned}
-\mathbb{E}_{p(y)}[\log \mathbb{E}_{p(x)}[p(y|x)]] &\leq -\mathbb{E}_{p(y)}\mathbb{E}_{p(x)}[\log p(y|x)] \\
&= \mathbb{E}_{p(x)}\mathbb{E}_{p(y)}[\log p(y|x)]
\end{aligned}
\tag{23}
$$

By applying this inequality to Equation (22), we obtain:

$$
\begin{aligned}
I(x;y) &= \mathbb{E}_{p(x,y)}[\log p(y|x)] - \mathbb{E}_{p(y)}[p(y)] \\
&= \mathbb{E}_{p(x,y)}[\log p(y|x)] - \mathbb{E}_{p(y)}[\log \mathbb{E}_{p(x)}[p(y|x)]] \\
&\leq \mathbb{E}_{p(x,y)}[\log p(y|x)] - \mathbb{E}_{p(x)}\mathbb{E}_{p(y)}[\log p(y|x)]
\end{aligned}
\tag{24}
$$

### 6.4 PROOF OF THEOREM 3.2

**Theorem 6.2** (Thereorem 3.2 restated). *Given two probability distributions $\mathbb{P}$, $\mathbb{Q}$. The Kullback Liebler Divergence admits the following dual representation:*

$$
D_{KL}(\mathbb{P}||\mathbb{Q}) = \sup_{T:\Omega \to \mathbb{R}} \mathbb{E}_{\mathbb{P}}[T] - \log \mathbb{E}_{\mathbb{Q}}[e^T],
\tag{25}
$$

*Proof.* The proof comprises two steps. Firstly, we prove the existence of the supremum in the dual representation. Subsequently, we demonstrate that this representation serves as the lower bound of the Kullback-Liebler Divergence.

**Lemma 1.** *There exist a function $T^* : \Omega \to \mathbb{R}$, such that:*

$$
D_{KL}(\mathbb{P}||\mathbb{Q}) = \mathbb{E}_{\mathbb{P}}[T^*] - \log \mathbb{E}_{\mathbb{Q}}[e^{T^*}]
\tag{26}
$$

*Proof.* We choose a function $T^* = \log \frac{\mathbb{P}}{\mathbb{Q}}$, then we have:

$$
\mathbb{E}_{\mathbb{P}}(T^*) - \log \mathbb{E}_{\mathbb{Q}}[e^{T^*}] = \mathbb{E}_{\mathbb{P}}\left[\log \frac{\mathbb{P}}{\mathbb{Q}}\right] - \log \mathbb{E}_{\mathbb{Q}}[e^{\log \frac{\mathbb{P}}{\mathbb{Q}}}]
\tag{27}
$$

$$
= D_{KL}(\mathbb{P}||\mathbb{Q}) - \log \mathbb{E}_{\mathbb{Q}}\left[\frac{\mathbb{P}}{\mathbb{Q}}\right]
\tag{28}
$$

$$
= D_{KL}(\mathbb{P}||\mathbb{Q}) - \log \int_{\Omega} \mathbb{Q}\frac{\mathbb{P}}{\mathbb{Q}} d\omega
\tag{29}
$$

$$
= D_{KL}(\mathbb{P}||\mathbb{Q}) - \log \int_{\Omega} \mathbb{P} d\omega
\tag{30}
$$

$$
= D_{KL}(\mathbb{P}||\mathbb{Q}) - \log 1
\tag{31}
$$

$$
= D_{KL}(\mathbb{P}||\mathbb{Q})
\tag{32}
$$

**Lemma 2.** *For any function $T : \Omega \to \mathbb{R}$, the following equality holds:*

$$
D_{KL}(\mathbb{P}||\mathbb{Q}) \geq \mathbb{E}_{\mathbb{P}}[T] - \log \mathbb{E}_{\mathbb{Q}}[e^T]
\tag{33}
$$

*Proof.* We define the probability density function $\mathbb{G}$ as:

$$
\mathbb{G} \triangleq \frac{\mathbb{Q}e^T}{\mathbb{E}_{\mathbb{Q}}[e^T]}
\tag{34}
$$

Note that $\mathbb{G}$ satisfies the non-negativity and the integral of its probability density function (PDF) over the input space equals 1:

$$\int_\Omega \mathbb{G}d\omega = \int_\Omega \frac{\mathbb{Q}e^T}{\mathbb{E}_\mathbb{Q}[e^T]}d\omega = \int_\Omega \frac{\mathbb{E}_\mathbb{Q}[e^T]}{\mathbb{E}_\mathbb{Q}[e^T]}d\omega = 1 \tag{35}$$

Then, we calculate the difference between the two sides of 42 to obtain:

$$D_{KL}(\mathbb{P}||\mathbb{Q}) - \mathbb{E}_\mathbb{P}[T] + \log\mathbb{E}_\mathbb{Q}[e^T] = \mathbb{E}_\mathbb{P}\left[\log\frac{\mathbb{P}}{\mathbb{Q}} - T\right] + \log\mathbb{E}_\mathbb{Q}[e^T] \tag{36}$$

$$= \mathbb{E}_\mathbb{P}\left[\log\frac{\mathbb{P}}{\mathbb{Q}e^T} + \log\mathbb{E}_\mathbb{Q}[e^T]\right] \tag{37}$$

$$= \mathbb{E}_\mathbb{P}\left[\log\frac{\mathbb{P}\mathbb{E}_\mathbb{Q}[e^T]}{\mathbb{Q}e^T}\right] \tag{38}$$

$$= \mathbb{E}_\mathbb{P}\left[\log\frac{\mathbb{P}}{\mathbb{G}}\right] \tag{39}$$

$$= D_{KL}(\mathbb{P}||\mathbb{G}) \geq 0 \tag{40}$$

Based on the Lemma 1 and Lemma 2, we show that by choosing $T^* = \log\frac{\mathbb{P}}{\mathbb{Q}}$, we obtain:

$$D_{KL}(\mathbb{P}||\mathbb{Q}) = \mathbb{E}_\mathbb{P}[T^*] - \log\mathbb{E}_\mathbb{Q}[e^{T^*}] \tag{41}$$

Additionally, for any function $T : \Omega \to \mathbb{R}$,

$$D_{KL}(\mathbb{P}||\mathbb{Q}) \geq \mathbb{E}_\mathbb{P}[T] - \log\mathbb{E}_\mathbb{Q}[e^T] \tag{42}$$

holds. Hence,

$$D_{KL}(\mathbb{P}||\mathbb{Q}) = \sup_{T:\Omega\to\mathbb{R}} \mathbb{E}_\mathbb{P}[T] - \log\mathbb{E}_\mathbb{Q}[e^T], \tag{43}$$

### 6.5 IMPLEMENTATION DETAILS

The trajectory denoise model $\Phi_{\textbf{TDM}}$ is implemented using a 3-layer Transformer with a feature dimension of 256 and the attention head is set to 4. The number of masked locations is set to 2 in our experiments. We empirically set the trade-off parameter $\beta$ to 0.01 and the margin $\Delta$ to 0.05. Additionally, we set the trade-off parameters $\alpha$, $\delta$, and $\gamma$ to 0.01, 1 and 0.01, respectively. For the Wavelet denoising method, we utilize the Daubechies wavelet to decompose the signals, and the level is set to 2. We employ the soft-threshold method, with a threshold value set to 0.2. Regarding the EMA method, we empirically determine the Weighted parameter to be 0.75. It is worth noting that these parameter selections are based on experiments aimed at ensuring optimal performance. All experiments are conducted on the PyTorch platform with 4 NVIDIA RTX3090 GPUs.

### 6.6 BROADER IMPACTS

This work addresses the challenge of trajectory prediction based on noisy observations. It enhances robustness against noise in the trajectory prediction task, benefiting various applications including autonomous driving, robotic navigation, and surveillance systems, thereby contributing to safer deployment.

### 6.7 TRAINING ALGORITHM OF NOISYTRAJ

We provide the training algorithm of NoisyTraj in the Algorithm 1.

### 6.8 DISCUSSSION AND LIMITATIONS

In this paper, we simplify the problem by assuming that only the observed trajectory is noisy, which is a reasonable assumption in certain scenarios. For example, when using an autonomous vehicle equipped with both cameras and LiDAR, we can treat camera-derived trajectories as noisy data and

---

**Algorithm 1:** Training Procedure of NoisyTraj

---

**Input:** Noisy observations $X_{obs}$, ground-truth future trajectories $Y_{fut}$. Four trade-off
hyper-parameters: $\alpha$, $\beta$, $\delta$ and $\gamma$.
**Output:** Network parameters: $\Phi_{\textbf{TDM}}$, $\Phi_{\textbf{TPB}}$, $\psi$, and $\phi$.
**Initialize:** Randomly initialize $\Phi_{\textbf{TDM}}$, $\Phi_{\textbf{TPB}}$, $\psi$, and $\phi$.
**while** *Model not converges* **do**

    Random mask the noisy observations using the mask vector: $X_{obs}^{mask} = X_{obs} \odot \mathcal{M}_{obs}$

    Obtain the trajectories $\hat{X}_{obs}^{mask} = \Phi_{\textbf{TDM}}(X_{obs}^{mask})$

    Calculate reconstruction loss $\mathcal{L}_{rec} = ||\hat{X}_{obs}^{mask} \odot (1 - \mathcal{M}_{obs}) - X_{obs} \odot (1 - \mathcal{M}_{obs})||_2$

    Input noisy observations to $\Phi_{\textbf{TDM}}$ for denoising: $\hat{X}_{obs} = \Phi_{\textbf{TDM}}(X_{obs})$

    Employ Mutual Information-based mechanism for further denoising:

$$\mathcal{L}_{MI} = \alpha \mathbb{E}_{p(X_{obs}, \hat{X}_{obs})}[\log q_\phi(\hat{X}_{obs}|X_{obs})] - \mathbb{E}_{p(X_{obs})}\mathbb{E}_{p(\hat{X}_{obs})}[\log q_\phi(\hat{X}_{obs}|X_{obs})]$$
$$- \sup_\psi \mathbb{E}_{p(\hat{X}_{obs}, Y_{fut})}[T\psi] + \log \mathbb{E}_{p(\hat{X}_{obs})p(Y_{fut})}[e^{T_\psi}]$$

    Obtain the future predictions based on denoised observations: $\{\hat{Y}_{fut}^k\}_{k=1}^K = \Phi_{\textbf{TPB}}(\hat{X}_{obs})$

    Obtain the future predictions based on noisy observation: $\{\tilde{Y}_{fut}^k\}_{k=1}^K = \Phi_{\textbf{TPB}}(X_{obs})$

    Calculate $d_{denoise}$ and $d_{noise}$:

$$d_{denoise} = \min_{1 \le k \le K} ||\hat{Y}_{fut}^k - Y_{fut}||_2, \quad d_{noise} = \min_{1 \le k \le K} ||\tilde{Y}_{fut}^k - Y_{fut}||_2$$

    Calculate $\mathcal{L}_{pred}$ and $\mathcal{L}_{rank}$ as

$$\mathcal{L}_{pred} = ||\hat{Y}_{fut}^{best} - Y_{fut}||_2 + ||\tilde{Y}_{fut}^{best} - Y_{fut}||, \quad \mathcal{L}_{rank} = \max(0, d_{denoise} - d_{noise} + \Delta)$$

    Optimizing $\mathcal{L} = \mathcal{L}_{pred} + \beta \mathcal{L}_{rank} + \delta \mathcal{L}_{rec} + \gamma \mathcal{L}_{MI}$ by gradient descent to update the $\Phi_{\textbf{TDM}}$
and $\Phi_{\textbf{TPB}}$.

**end**

---

LiDAR-derived trajectories as clean ground-truth for training. Once the model is trained on this data, it can be deployed on a vehicle equipped with only cameras. This camera-only approach is adopted by top industry Tesla to design the Autopilot system, which has been successfully deployed in real-world scenarios [2].

While this work focuses on addressing trajectory prediction based on noisy observed trajectories, it is important to acknowledge that the collected future ground-truth trajectories may also be contaminated with noise. In such cases, the proposed mutual information-based denoising mechanism may not be effective, as NoisyTraj assumes the future trajectories are noise-free and uses them as additional information for denoising the observations. Future research could explore methods for predicting future trajectories based on both noisy observations and noisy future ground-truths.

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
