# OpenReview forum: "NoisyTraj: Robust Trajectory Prediction with Noisy Observations"
_ICLR.cc/2025/Conference — ICLR 2025 Conference Withdrawn Submission_

### Official Review · Reviewer_fxit · 2024-10-21

**Soundness:** 2
**Presentation:** 2
**Contribution:** 3
**Rating:** 5
**Confidence:** 5

**Summary:**

This paper addresses the challenge of noise in real-world trajectory prediction tasks, which arises from errors in sensors, detection, and tracking processes. The authors propose a noise-agnostic approach, NoisyTraj, capable of handling arbitrary types of noisy observations. NoisyTraj employs three key strategies: a mutual information-based mechanism for denoising, a reconstruction task to preserve trajectory structure, and a ranking loss to ensure superior performance with denoised data. The method has been evaluated on the ETH/UCY and Stanford Drone datasets (SDD), where it demonstrated significant improvements over baseline models in predicting trajectories from noisy observations.

**Strengths:**

- The concept of denoising historical trajectory data is innovative.
- The proposed denoising method is simple to implement and can be integrated into existing models as a plug-and-play component.
- The model shows promising performance in the experimental conditions set by the authors.

**Weaknesses:**

- The relationship between the three proposed improvements, particularly the ranking loss, lacks a clear intrinsic connection. While the paper states that the ranking loss enhances performance, it does not provide a detailed explanation of why this is the case. It would be beneficial for the authors to include ablation studies on the hyperparameters of the loss function and clarify why adding ranking loss leads to better results compared to designing a prediction loss that only accounts for denoised prediction results.
- The experimental setup is insufficient. The two baselines used are overly simplistic, limiting the ability to assess the model’s effectiveness fully. Including the prediction results from the historical trajectory data before adding noise in all experiments would strengthen the comparisons.
- The paper lacks sufficient detail on the model’s training process. Specifically, whether the training dataset was also subjected to noise is unclear. Additionally, the limitation that the model training must be tailored to a specific trajectory prediction backbone, and that freezing the trajectory prediction backbone parameters may degrade performance, should be addressed in the main text.

**Questions:**

The paper introduces noise into the original dataset to create noisy historical trajectory data. If the original data is considered noise-free, should the experiment include a similarity comparison between the denoised input and the pre-noise data? Alternatively, if the original data itself is considered noisy, should the experiment demonstrate that the model’s predictions using denoised data outperform those using the data before noise was added?

---

### Official Review · Reviewer_gjxC · 2024-10-27

**Soundness:** 2
**Presentation:** 4
**Contribution:** 4
**Rating:** 5
**Confidence:** 3

**Summary:**

Conventional trajectory prediction assumes that clean trajectories are provided; however, actual trajectory data contains noise due to errors from sensors, detection, and tracking processes.
These noises can degrade prediction performance, so the authors propose a mutual information-based mechanism to denoise the original noisy observations.
This approach optimizes the generated trajectories to resemble clean trajectory patterns while diverging from noisy ones.
To preserve structural information when using the MI-based method, they also introduce a masking-based reconstruction loss.
Additionally, under the assumption that denoised trajectories lead to better prediction performance, a ranking loss is proposed.
Since the proposed method does not assume a specific type of noise, it is robust against various forms of noise, and its effectiveness is demonstrated on the ETH-UCY and SDD datasets, under different noise types.

**Strengths:**

* The proposed concepts and their underlying motivations—mutual information-based mechanism, reconstruction loss, and ranking loss—sound solid.
* The mathematical modeling of minimizing the upper bound of the mutual information mechanism is well-structured, and its effectiveness demonstrated in the ablation study shows incremental improvement.
* The experimental results are comprehensive. The method is tested across multiple datasets and shows superior performance across all metrics. Demonstrating generalizability across various types of noise is also a strong point.

**Weaknesses:**

* The recent paper (https://arxiv.org/abs/2312.15906) addresses transfer learning, but focuses specifically on making robust predictions in the presence of noisy trajectories within datasets. Since it deals with actual noise arising from errors in sensors, detection, and tracking processes, a comparison with this paper seems necessary.
* The assumption that $Y_{fut}$ is clean sounds somewhat unrealistic. Even for trajectories obtained via Lidar sensors, the mentioned errors from sensors, detection, and tracking processes are inevitable, although they might be less than those from camera-driven trajectories. It would be more convincing to show experimental results in settings where such two real-world different noise levels exist, for example, use clear trajectory as real Lidar-driven trajectories and noisy trajectory as real camera-driven trajectories.
* Typo: line 178, change X^ to X^$_{obs}$.

**Questions:**

My major concern is:
If obtaining a clean trajectory is possible, then it should also be possible to obtain a clean historical trajectory. In that case, instead of the proposed mutual information-based method, wouldn’t it be sufficient to use an L2 loss to train a denoiser to restore the clean observation trajectory from the noisy one? This makes me wonder if the content in Section 3.3.1 would still be necessary.

I believe that the presentation and contributions of this paper are meaningful. However, for the proposed method in this paper to be practical, the assumption made—obtaining a clean future trajectory is feasible—needs to be convincing. I am skeptical about this aspect, which is why I have given a lower score.

---

### Official Review · Reviewer_xtpN · 2024-11-02

**Soundness:** 3
**Presentation:** 3
**Contribution:** 3
**Rating:** 8
**Confidence:** 4

**Summary:**

The manuscript tackles trajectory prediction with noisy inputs by proposing a denoising module with two main steps: i) generating clean trajectories via mutual information maximization, and ii) reconstructing noisy trajectories from masked segments to retain structure. The authors also introduced a ranking loss to produce more accurate prediction. The proposed denoising module is a plug-and-play module and is validated on ETH/UCY and Stanford Drone datasets (SDD) using different existing backbones.

**Strengths:**

1. The manuscript is well-organized and easy to follow.

2. The use of mutual information to denoise input trajectories is both intuitive and effective. The reconstruction loss is well-motivated.

3. Experimental results highlight the capability and generalizability of the proposed plug-and-play trajectory denoising module.

4. The experiment section and the ablations in the appendix is very convincing.

**Weaknesses:**

The task is new and challenging, there is not much real data to use for validation. This reviewer appreciate the author's efforts but maybe using additive gaussian noise is not the best way to simulate the realistic sensor input. It may cause model overfitting of certain types of the noise.

**Questions:**

N/A

---

### Official Review · Reviewer_5VCh · 2024-11-04

**Soundness:** 2
**Presentation:** 3
**Contribution:** 3
**Rating:** 6
**Confidence:** 3

**Summary:**

The paper proposes NoisyTraj to deal with noisy observations in trajectory prediction tasks. NoisyTraj is noise-agnostic and leverages a mutual information based mechanism to filter noise from the observation in the Trajectory Denoise Model (TDM) before feeding the denoised observations into the Trajectory Prediction Backbone (TPB), which predicts the future trajectory. Any trajectory prediction model can trivially be chosen as the TPB.
The optimization maximizes the mutual information (MI) between produced and future trajectories, and minimizes the MI between produced and input (noisy) trajectories. In addition, reconstruction losses based on random masking and a trajectory ranking loss (and a trajectory prediction loss) are applied.
The authors validate the method on ETH/UCY and SDD datasets, to which they add various synthetic noises. Results show superior performance over non-denoising and simple denoising (Wavelet, EMA) baselines, even when train and test noise are varied.

**Strengths:**

* Beats baselines: NoisyTraj beats all baselines (no denoising, Wavelet denoising, EMA denoising) across all datasets (ETH/UCY and SDD) and various noise settings.
* No noise-free degradation: NoisyTraj performs equally well as the no denoising baseline when no noise is added (exists). In addition, NoisyTraj might generalize to different train / test noises.
* Ablations: Ablation studies are conducted and show that all losses contribute to NoisyTraj's performance
* Well presented: The work is well presented and easy to follow.

**Weaknesses:**

* Limited set of applications: The work tackles a simplified problem, in which the future ground truth data is assumed to be noise-free, while only the input observations are noisy. In general trajectory prediction tasks, e.g. in autonomous driving, this setup does not hold true. The authors give only one example where this method could be helpful: Use camera-only as inputs and camera + LiDAR as ground truth. However, even in this example camera + LiDAR are not noise-free as assumed in the paper but might have a lower / different noise than the input data. It would be of significant value if such a setup, where the ground truth data is not noise-free would be evaluated to confirm whether NoisyTraj still works well or whether it needs to be adapted. Furthermore, this very restrictive assumption should be mentioned at the beginning of the paper (if I'm not mistaken, it's only first mentioned in Section 3).
* Evaluation with higher noise values: /sigma=0.2 and 0.4 are evaluated but both noise levels are quite similar. How does NoisyTraj perform with larger noises, e.g. /sigma=2? From the results in Table 2, it also seems that the gap between NoisyTraj and the baselines is not widened besides the 2x noise increase. More data and analysis would be helpful here.
* Generalizability: The results in Table 5 actually show that the gap between the methods narrows when different train / test noises are used (while NoisyTraj still outperforms the baselines). Given that the baselines are rather weak (no good baselines seem to exist), it's unclear whether NoisyTraj truly generalizes well.
Synthetic noise: The paper would be stronger if validation also covered real noise, e.g. from autonomous driving tasks. Real noisy observations with noise-free ground truth data is generally difficult to obtain though. One work that I'm aware of that analyzed the noise of a detection + tracking pipeline is https://arxiv.org/pdf/2004.01288, but they did not publish the dataset.

**Questions:**

* How does NoisyTraj perform with more noisy data, e.g. /sigma=2?
* How does NoisyTraj perform when the ground truth data also contains some (likely less) noise?
* Can you evaluate NoisyTraj on real noised trajectories?

---

### Official Review · Reviewer_WDuf · 2024-11-09

**Soundness:** 2
**Presentation:** 3
**Contribution:** 2
**Rating:** 3
**Confidence:** 5

**Summary:**

The paper presents a method called NoisyTraj for trajectory prediction upon noisy observations. The key idea of the method is to learn a trajectory denoise model using the mutual information between trajectories. This method can be applied to different baseline prediction models. The effectiveness of the method is verified on two popular human trajectory prediction benchmarks.

**Strengths:**

The paper is well written. The motivation, method, and experiments are well presented.

**Weaknesses:**

1. The major weakness of this paper is that the problem formulation does not align with real-world scenarios. The paper assumes that observed trajectories are corrupted by noise, and the authors add specific noise (e.g., Gaussian) to clear observations to simulate noisy observations. However, in practical applications, trajectory observation noise and errors are much more complex. In detection, not only do bounding boxes have noise, but there are also numerous false positives and false negatives. In subsequent tracking, bounding boxes belonging to the same subject need to be associated across different timestamps. Incorrect associations may link one subject's box at time T to another subject's box at T+1. Therefore, errors in the association process can result in tracked trajectories that significantly deviate from ground truth trajectories. The paper's assumption that noisy observations only stem from imprecise observed positions is fundamentally different from actual noisy/erroneous observations obtained from tracking, making it far from practical applications.

2. Another major weakness is the omission of several important prior works that have already explored trajectory prediction with noisy observations:

[a] Yu, Rui, and Zihan Zhou. "Towards robust human trajectory prediction in raw videos." 2021 IEEE/RSJ International Conference on Intelligent Robots and Systems (IROS). IEEE, 2021.

[b] Weng, Xinshuo, Boris Ivanovic, and Marco Pavone. "MTP: Multi-hypothesis tracking and prediction for reduced error propagation." 2022 IEEE Intelligent Vehicles Symposium (IV). IEEE, 2022.

[c] Weng, Xinshuo, et al. "Whose track is it anyway? improving robustness to tracking errors with affinity-based trajectory prediction." Proceedings of the IEEE/CVF Conference on Computer Vision and Pattern Recognition. 2022.

[d] Zhang, Pu, et al. "Towards trajectory forecasting from detection." IEEE Transactions on Pattern Analysis and Machine Intelligence 45.10 (2023): 12550-12561.

Different from NoiseTraj in this paper that focuses solely on position-based noise, these prior works consider the comprehensive impact of tracking-related issues on prediction, For instance, paper [a] provides an analysis of how various tracking problems (including noisy tracks, missed detections, spurious tracks, and ID switches) affect trajectory prediction performance. NoiseTraj could serve as a trajectory smoothing component, similar to how the Holt-Winters method is employed in [a].

3. Due to the strong coupling between tracking and prediction, addressing prediction upon noisy observations requires considering both tracking and prediction. However, this paper only focuses on prediction without considering tracking. Artificially adding noise to clear observations creates an artificial problem setting. Moreover, the evaluation metrics cannot simply adopt ADE and FDE from prediction upon clean observations. Since observed trajectories might be incorrect (rather than just having inaccurate positions), evaluation metrics need to consider both tracking and prediction. For example, ADE-over-recall curves are used in [a]&[e].

[e] Weng, Xinshuo, et al. "Inverting the pose forecasting pipeline with SPF2: Sequential pointcloud forecasting for sequential pose forecasting." Conference on robot learning. PMLR, 2021.

4. Figure 3 demonstrates that artificially noisy trajectories do not match the patterns of real-world noisy tracklets. The offset in the noisy (blue) trajectory exceeds several person-distances. In practical tracking associations, these points would never be associated with the same trajectory. For reference on normal noise patterns, see Fig. 5 in [a] (IROS'21).

5. In Section 4.2, the noise added to the SDD dataset is measured in meters, while the evaluation is in pixels. These experimental details need clarification.

6. Expression issues:
a) Line 360: "blue" arrow should refer to the "green" arrow in Figure 2?
b) Line 451: "Kalman" should refer to "Wavelet"?

**Questions:**

To address the identified weaknesses, the following suggestions are provided:
1. Discuss how the proposed method could be extended to handle realistic noise or error scenarios, such as false positives/negatives and association errors.
2. Provide a comparison of the proposed method with prior works.

To further improve the paper, the following suggestions are provided:

3. Conduct additional experiments using realistic noise/error data instead of synthetic noise.
4. Clarify the scope of the proposed method and discuss its limitations in real-world scenarios.
5. Explore potential ways to incorporate NoiseTraj with prior works.
6. Discuss how the proposed method could be integrated with or extended to include tracking components in the future.

---

### Note · Authors · 2024-11-14

I have read and agree with the venue's withdrawal policy on behalf of myself and my co-authors.